# Distributed quantum sensing of multiple phases with fewer photons

**Dong-Hyun Kim**[1,2,10], **Seongjin Hong**[3,10], **Yong-Su Kim**[1,4], **Yosep Kim**[1,5], **Seung-Woo Lee**[1], **Raphael C. Pooser**[6], **Kyunghwan Oh**[2], **Su-Yong Lee**[7,8], **Changhyoup Lee**[9] **& Hyang-Tag Lim**[1,4] ✉

Distributed quantum metrology has drawn intense interest as it outperforms the optimal classical counterparts in estimating multiple distributed parameters. However, most schemes so far have required entangled resources consisting of photon numbers equal to or more than the parameter numbers, which is a fairly demanding requirement as the number of nodes increases. Here, we present a distributed quantum sensing scenario in which quantum-enhanced sensitivity can be achieved with fewer photons than the number of parameters. As an experimental demonstration, using a two-photon entangled state, we estimate four phases distributed 3 km away from the central node, resulting in a 2.2 dB sensitivity enhancement from the standard quantum limit. Our results show that the Heisenberg scaling can be achieved even when using fewer photons than the number of parameters. We believe our scheme will open a pathway to perform large-scale distributed quantum sensing with currently available entangled sources.

Quantum metrology achieves enhanced sensitivity for estimating unknown parameters beyond the standard quantum limit (SQL), which is the sensitivity bound attainable by exploiting classical resources[1–6]. Recently, developments in quantum metrology have been directed toward distributed quantum sensing[7–15]. Unlike conventional quantum sensing, which estimates single or multiple parameters at a single location, the goal of distributed quantum sensing is to estimate the linear combination of multiple unknown parameters distributed among distant nodes[7,8,16]. Distributed quantum sensing can be used as a quantum sensor network, and it has been known to be useful for various applications such as local beam tracking, and global-scale clock synchronization[17,18]. Numerous efforts have been thus made to achieve quantum-enhanced sensitivity for distributed quantum sensing by utilizing quantum resources[19–22].

Various strategies have been proposed for distributed multiple-phase sensing by using entangled probe states among distributed sensors. In continuous-variable (CV) quantum metrology, it has been reported that an entangled CV state can achieve sensitivity beyond what is attainable with separable states[19,20,23]. On the other hand, in discrete-variable (DV) quantum metrology, it was theoretically proven that the Heisenberg scaling (HS) could be achieved by utilizing both mode-entangled and particle-entangled (MePe) states[16], and an experimental demonstration of estimating an average of three unknown phases was realized in ref. 21. Moreover, a demonstration of estimating an average of two unknown phases over a fiber distance of 10 km was realized in ref. 22. However, in order to estimate multiple unknown phases using the MePe states, the number of photons in the MePe states should be equal to or larger than the number of unknown phases. For example, to estimate the average of three phases, six-photon MePe states were used in ref. 21, and two-photon MePe states were used to estimate the average of two phases in ref. 22. Therefore, the practical scalability of distributed quantum sensing to estimate

[1]Center for Quantum Information, Korea Institute of Science and Technology (KIST), Seoul 02792, Korea. [2]Department of Physics, Yonsei University, Seoul 03722, Korea. [3]Department of Physics, Chung-Ang University, Seoul 06974, Korea. [4]Division of Nanoscience and Technology, KIST School, Korea University of Science and Technology, Seoul 02792, Korea. [5]Department of Physics, Korea University, Seoul 02841, Korea. [6]Oak Ridge National Laboratory, Oak Ridge, TN 37831, USA. [7]Emerging Science and Technology Directorate, Agency for Defense Development, Daejeon 34186, Korea. [8]Weapon Systems Engineering, ADD School, University of Science and Technology, Daejeon 34060, Korea. [9]Korea Research Institute of Standards and Science, Daejeon 34113, Korea. [10]These authors contributed equally: Dong-Hyun Kim and Seongjin Hong. ✉e-mail: hyangtag.lim@kist.re.kr

multiple unknown phases has been limited by the difficulty associated with generating multi-photon entangled states[24].

In this work, we propose a distributed quantum sensing protocol that achieves quantum-enhanced sensitivity even when the number of photons used is less than the number of unknown phases to estimate. We also experimentally demonstrate distributed quantum-enhanced detection of the average of four phases located in distant nodes over 3 km away from the central node with two-photon entangled states. To this end, we distribute the prepared two-photon entangled state among four nodes. Then the probe states undergo phase encoding at each node, which is separated by 3 km away from the central node through optical fiber spools, and are measured by four local measurements. We estimate an average of four unknown phases with the maximum likelihood estimator (MLE), and achieved a sensitivity beyond the SQL. We believe that our results provide a useful platform to investigate a distributed quantum sensor network.

## Results

Let us begin by introducing a general scenario of distributed quantum sensing to estimate the spatially distributed $d$ multiple phases as shown in Fig. 1[8,16]. There are $d$ unknown phases $\boldsymbol{\phi} = (\phi_1, \phi_2, ..., \phi_d)$, and they are distributed in different locations. In a distributed quantum sensing scenario, the goal is to estimate a linear global function of $\hat{\phi} = \boldsymbol{\alpha}^T \boldsymbol{\phi}$, where $\boldsymbol{\alpha} = (\alpha_1, \alpha_2, ..., \alpha_d)$ and $\sum_{j=1}^{d} |\alpha_j| = 1$. Then the unitary phase evolution is given by $\hat{U}(\boldsymbol{\phi}) = \exp(-i \sum_{j=1}^{d} \hat{H}_j \phi_j) = \exp(-i \hat{\mathbf{H}} \cdot \boldsymbol{\phi})$, where $\hat{\mathbf{H}} = (\hat{H}_1, ..., \hat{H}_d)$ denotes the local Hamiltonians. The initial probe state $|\Psi\rangle$ interacts with phase encoding $\hat{U}(\boldsymbol{\phi})$ and evolves into $\hat{U}(\boldsymbol{\phi})|\Psi\rangle$. Finally, it is detected by a set of projectors $\{\hat{\Pi}_l\}$, and it gives a corresponding probability set $\{P_l\}$ for a given $\boldsymbol{\phi}$. Then, we can estimate $\hat{\phi}_{\text{est}}$ from $\{P_l\}$ using a proper estimator, for instance, MLE[21,22,25]. Here, the uncertainty bound for the estimation of $\hat{\phi}$ can be described as follows:

$$\Delta^2 \hat{\phi} \equiv \langle (\hat{\phi}_{\text{est}} - \hat{\phi})^2 \rangle \geq \frac{(\boldsymbol{\alpha}^T \boldsymbol{\alpha})^2}{\mu \boldsymbol{\alpha}^T \mathbf{F} \boldsymbol{\alpha}}, \quad (1)$$

which is known as the weak form of Cramer-Rao bound (CRB), $\mathbf{F}$ denotes the Fisher information matrix with elements $F_{(j, k)} = \sum_l (1/P_l)$

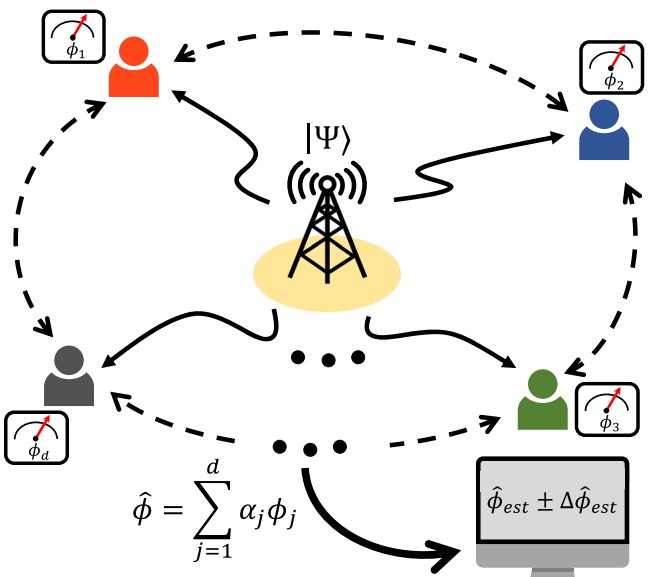

**Fig. 1 | Scheme for estimating spatially distributed multiple phases.** Multiple unknown phases $\boldsymbol{\phi} = \{\phi_1, \phi_2, ..., \phi_d\}$ are located in each node. The goal of distributed sensing is to estimate the linear combination of the multiple unknown phases, $\hat{\phi}$, with a quantum-enhanced sensitivity.

$(\partial P_l / \partial \phi_j)(\partial P_l / \partial \phi_k)$, and $\mu$ denotes the number of measurements[16]. Hereafter, we will simplify the description with $\mu = 1$ for all the presented theoretical schemes.

To achieve the HS ($\Delta^2 \hat{\phi} = 1/N^2$), the MePe state, which was proposed in refs. 16 and [21], has the form of

$$|\Psi_{\text{MePe}}\rangle = \frac{1}{\sqrt{2}} \left( \otimes_{j=1}^{d} |H_j\rangle^{\otimes N/d} + \otimes_{j=1}^{d} |V_j\rangle^{\otimes N/d} \right), \quad (2)$$

where $H(V)$ represents a horizontal (vertical) polarization, $N$ denotes the total number of photons, $d$ denotes the number of unknown phases (i.e., number of nodes), and $N/d$ is the number of photons in each node $j = 1, 2, ..., d$. In Eq. (2), the photon number $N$ should be equal to or larger than the number of unknown phases $d$. The MePe state requires the photon number $N = m \times d$ ($m$ is a positive integer). Thus, to achieve the HS using a MePe state in distributed sensing, at least $d$-photon GHZ-like entangled states are required. However, a key challenge in the approach is the difficulty in generating $d$-photon MePe states for large $d$[24].

Here, instead, we consider another probe state that can achieve the HS without the constraint $N \geq d$. The probe state we propose has the form of

$$\left| \Psi_d^N \right\rangle = \frac{1}{\sqrt{2d}} \sum_{j=1}^{d} \left( |H_j\rangle^{\otimes N/2} |H_{j+1}\rangle^{\otimes N/2} + |V_j\rangle^{\otimes N/2} |V_{j+1}\rangle^{\otimes N/2} \right), \quad (3)$$

where $d + 1 \equiv 1$ since node $d$ is adjacent to node 1 as shown in Fig. 1, for instance, $|H_d\rangle |H_{d+1}\rangle \equiv |H_d\rangle |H_1\rangle$. Equation (3) corresponds to a coherent superposition of the $N$-photon two-mode polarization-entangled states in which $N/2$ photons are distributed in two adjacent nodes, respectively. In this strategy, the photon number in each node is $N/2$, so the only condition of our probe state is that $N$ is an even number. It means that our probe state can be used to estimate $d$ unknown phases with $N$ photons even though $N < d$. By adding a beam splitter network (BSN) to multiplex the state, we are able to simultaneously probe multiple paths. In our experiment, we consider a two-photon probe state $\left| \Psi_4^2 \right\rangle$ to estimate four unknown phases, i.e., $N = 2$ and $d = 4$.

For demonstration purposes, we experimentally estimate an average of four spatially distributed phases $\hat{\phi} = \sum_{j=1}^{4} \phi_j / 4$ (see Fig. 2a), and $\phi_j$ corresponds to the phase encoding at node $j$. Note that four sensors are distributed 3 km away from the central node, respectively. To this end, we prepared the two-photon polarization entangled probe states in Eq. (3) with $N = 2$ and $d = 4$ distributed in four nodes as shown in Fig. 2b. Firstly, the polarization Bell state $|\Phi_{a,b}\rangle = (|H_a H_b\rangle + |V_a V_b\rangle)/\sqrt{2}$ was generated via spontaneous parametric down-conversion (SPDC) process from a 10-mm-thick type-II periodically-poled KTiOPO$_4$ (PPKTP) crystal, which is located at the Sagnac-interferometer[26,27]. Then, the generated Bell state was split to distribute the quantum states among four nodes by the BSN consisting of two 50/50 beam splitters as shown in Fig. 2b. The process of preparing $\left| \Psi_4^2 \right\rangle$ can be described as follows:

$$|\Phi_{a,b}\rangle = \frac{1}{\sqrt{2}} (|H_a H_b\rangle + |V_a V_b\rangle)$$
$$\xrightarrow{\text{BSN}} \left| \Psi_4^2 \right\rangle = \frac{1}{2} (|\Phi_{1,2}\rangle + |\Phi_{2,3}\rangle + |\Phi_{3,4}\rangle + |\Phi_{4,1}\rangle)$$
$$\xrightarrow{\boldsymbol{\phi}} \hat{U}(\boldsymbol{\phi}) \left| \Psi_4^2 \right\rangle = \frac{1}{2} [(|H_1 H_2\rangle + e^{i(\phi_1 + \phi_2)} |V_1 V_2\rangle)$$
$$+ (|H_2 H_3\rangle + e^{i(\phi_2 + \phi_3)} |V_2 V_3\rangle)$$
$$+ (|H_3 H_4\rangle + e^{i(\phi_3 + \phi_4)} |V_3 V_4\rangle)$$
$$+ (|H_4 H_1\rangle + e^{i(\phi_4 + \phi_1)} |V_4 V_1\rangle)], \quad (4)$$

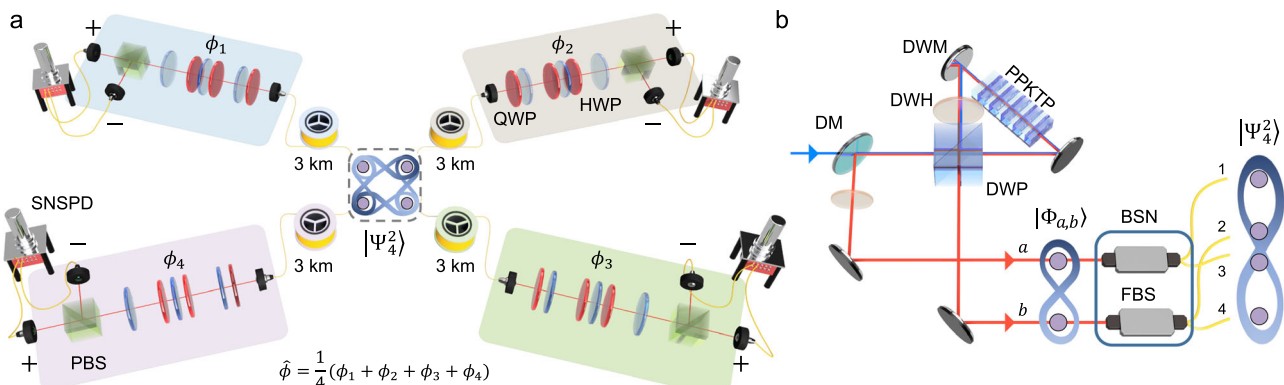

**Fig. 2 | Experimental scheme. a** Experimental setup for demonstrating our distributed quantum sensing scenario. The entangled probe state $|\Psi_4^2\rangle$ at the central node is sent to each node via 3 km fiber spools. Then, $|\Psi_4^2\rangle$ undergoes phase shifts at each node with a combination of two QWPs, and a HWP. In each node, $\sigma_x$ measurement is performed using a set of a HWP with an optic axis angle of 22.5° and a PBS. Then, the two-photon coincidence counts are measured using superconducting nanowire single-photon detectors (SNSPDs). **b** A polarization entangled state $|\Phi_{a,b}\rangle$ is generated from a Sagnac-interferometer-based Bell source[26,27]. Then, $|\Psi_4^2\rangle$ is prepared by using a BSN consisting of two FBSs. QWP: quarter waveplate; HWP: half waveplate; PBS: polarizing beam splitter; DM: dichroic mirror; DWP: dual wavelength PBS; DWM: dual wavelength mirror; DWH: dual wavelength HWP; PPKTP: periodically-poled KTiOPO; BSN: beam splitter network; FBS: 50/50 fiber beam splitter. The detailed information on our experimental setup is provided in Supplementary Note.

where the subscript denotes the node, for example, $|\Phi_{j,k}\rangle$ denotes the Bell states $(|H_j H_k\rangle + |V_j V_k\rangle)/\sqrt{2}$ between the node $j$ and the node $k$. Then, $|\Psi_4^2\rangle$ can be interpreted as a superposition of four Bell states between the two adjacent nodes such as $|\Phi_{1,2}\rangle$, $|\Phi_{2,3}\rangle$, $|\Phi_{3,4}\rangle$, and $|\Phi_{4,1}\rangle$. Then, the probe state $|\Psi_4^2\rangle$ evolved into $\hat{U}(\boldsymbol{\phi})|\Psi_4^2\rangle$ after the phase encoding. The phase encoding was realized by a set of quarter waveplate (QWP), half waveplate (HWP), and QWP located at each node, as shown in Fig. 2a[26,27].

Then $\hat{U}(\boldsymbol{\phi})|\Psi_4^2\rangle$ was measured by projective measurement on the $\sigma_x$ basis at each node[21,22]. The $\sigma_x$ measurement was performed with a combination of an HWP at 22.5° from the optic axis and a polarizing beam splitter (PBS), and the coincidence measurement in the $\sigma_x$ basis was performed in each node. Then, it gives a corresponding probability set $\{P_l\}$, and there are sixteen probabilities of $\{P_{jk}^{++}, P_{jk}^{+-}, P_{jk}^{-+}, P_{jk}^{--}\}$ where the superscripts of + and − correspond to the outcomes of the $\sigma_x$ measurement, and $jk$ denotes the nodes and depending on $\phi_j$ and $\phi_k$, i.e., $\{jk\} = \{12, 23, 34, 41\}$. For example, $P_{12}^{+-}$ denotes the outcome probability of two-photon coincidence events between + in node 1 and − in node 2. The outcome probabilities in our experiment can be written as

$$P_{jk}^{++} = P_{jk}^{--} = \frac{1 + V_{jk}^{\pm\pm} \cos(\phi_j + \phi_k)}{16},$$
$$P_{jk}^{+-} = P_{jk}^{-+} = \frac{1 - V_{jk}^{\pm\mp} \cos(\phi_j + \phi_k)}{16}, \tag{5}$$

where $V_{jk}^{\pm\pm}$ ($V_{jk}^{\pm\mp}$) is the visibility of $P_{jk}^{\pm\pm}$ ($P_{jk}^{\pm\mp}$). The sensitivity bound of our probe state can be calculated using Eq. (5) and reach the HS beyond the SQL. See Methods for the detailed calculation.

Our experimental results on $\{P_{12}^{++}, P_{12}^{+-}, P_{12}^{-+}, P_{12}^{--}\}$ by scanning $\phi_1$ and $\phi_2$ are shown in Fig. 3, and their averaged visibilities are obtained to be 0.955, 0.981, 0.970, and 0.945, respectively. Note that $P_{12}, P_{23}, P_{34}$ and $P_{41}$ are the outcome probabilities with respect to $\{\phi_1, \phi_2\}$, $\{\phi_2, \phi_3\}$, $\{\phi_3, \phi_4\}$, and $\{\phi_4, \phi_1\}$, respectively. We provide our experimental results on other probabilities in Supplementary Notes. Then, we estimate an average of unknown four phases, i.e., $\hat{\phi}$, using MLE[21,22]. To obtain the sensitivity for estimating $\hat{\phi}$, we calculated the standard deviation of the estimated average of four phases, i.e., $\Delta\hat{\phi}_{est}$. To this end, the sixteen probabilities $\{P_l\}$ were obtained by scanning $\phi_1$ when $\phi_2, \phi_3$, and $\phi_4$ are fixed at $\phi_2 \simeq \pi/2$ and $\phi_3 \simeq \phi_4 \simeq 0$. The experimental probabilities were calculated from the measured post-selected coincidence events with $\mu \simeq 367$ as shown in Fig. 4a, b. In addition, the expected limit was

obtained by calculating Eq. (1) using the sixteen probabilities with experimentally obtained non-ideal visibilities, and the SQL and the HS are also drawn by calculating $1/(\sqrt{\mu N})$ and $1/(\sqrt{\mu} N)$, respectively, with $\mu \simeq 367$ in Fig. 4c. See Methods for error analysis on our experimental results. Note that the error bars on the standard deviation in Fig. 4c lie below the HS. It is mainly from the mismatch between experimental results and theoretically calculated probability functions in Figs. 4a, b, and we attribute this to fluctuation on phases due to a few km long optical fibers (~3 km) and the fact that the limited number of samples are used for estimation ($\mu \simeq 367$)[22]. Our experimental results clearly show that our strategy can achieve sensitivity beyond the SQL, and this can be further improved up to the HS with better visibility. We can also see that the obtained standard deviations are close to the HS in Fig. 4c.

In our experiment, we used a post-selection technique and did not include experimental imperfections such as photonic losses and lack of high-efficiency photon number resolving detectors. However, verifying the proof-of-concept of quantum enhancement is not affected by the post-selection technique, which has been used in most experiments for quantum-enhanced parameter metrology[21,25–29]. Moreover, it is possible to achieve sensitivity beyond the SQL by using state-of-the-art high-efficiency photon number resolving detectors[22,30,31]. See Supplementary Note for an analysis that takes into account all losses in our experiment setup.

We now consider a generalization of our scheme with Eq. (3) to $N$-photon and $d$ unknown phases to estimate an average phase $\hat{\phi} = \sum_{j=1}^{d} \phi_j/d$ as shown in Fig. 5. The extended probe state in Eq. (3) can be described differently as follows:

$$\left|\Psi_d^N\right\rangle = \frac{1}{\sqrt{d}}\left(\left|\Phi_{1,2}^N\right\rangle + \left|\Phi_{2,3}^N\right\rangle + \ldots + \left|\Phi_{d-1,d}^N\right\rangle + \left|\Phi_{d,1}^N\right\rangle\right). \tag{6}$$

For example, $\left|\Phi_{1,2}^N\right\rangle$ denotes the $N$-photon two-mode polarization-entangled states between node 1 and node 2, i.e., $\left|\Phi_{1,2}^N\right\rangle = (|H_1\rangle^{\otimes N/2}|H_2\rangle^{\otimes N/2} + |V_1\rangle^{\otimes N/2}|V_2\rangle^{\otimes N/2})/\sqrt{2}$, meaning that $N/2$ photons are sent to each node, respectively[32,33]. Here, $N$ is assumed to be even. Then one can calculate the Fisher information matrix for $\left|\Psi_d^N\right\rangle$ as follows:

$$F_{(j,k)} = \begin{cases} \frac{N^2}{2d} & \text{if } j = k \\ \frac{N^2}{4d} & \text{if } j = k \pm 1 \\ 0 & \text{otherwise} \end{cases} \tag{7}$$

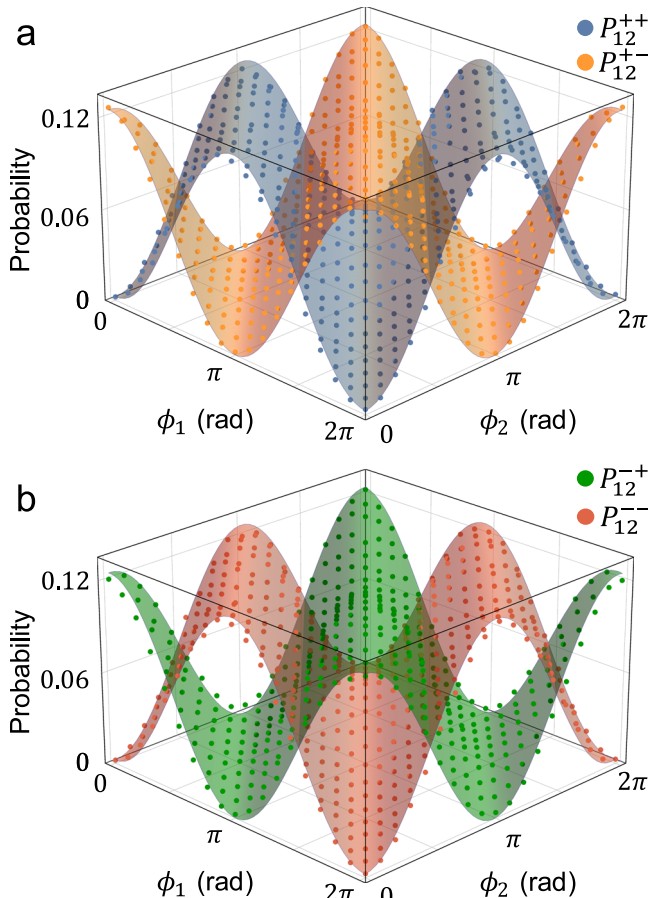

**Fig. 3 | Experimental results on interference fringes between the node 1 and the node 2.** The distance between the two nodes is 6 km (fiber spool). **a, b** Two-photon coincidence counts are obtained when the phase encoding $\hat{\phi} \in [0, 2\pi]$. Each surface represents a theoretical model $P_{12}^{++}, P_{12}^{+-}, P_{12}^{-+}$, and $P_{12}^{--}$ in Eq. (5), respectively, and dots correspond to our experimental data. Note that the error bars are smaller than the markers. See Supplementary Note for the experimental interference fringe results that are obtained by scanning all phases.

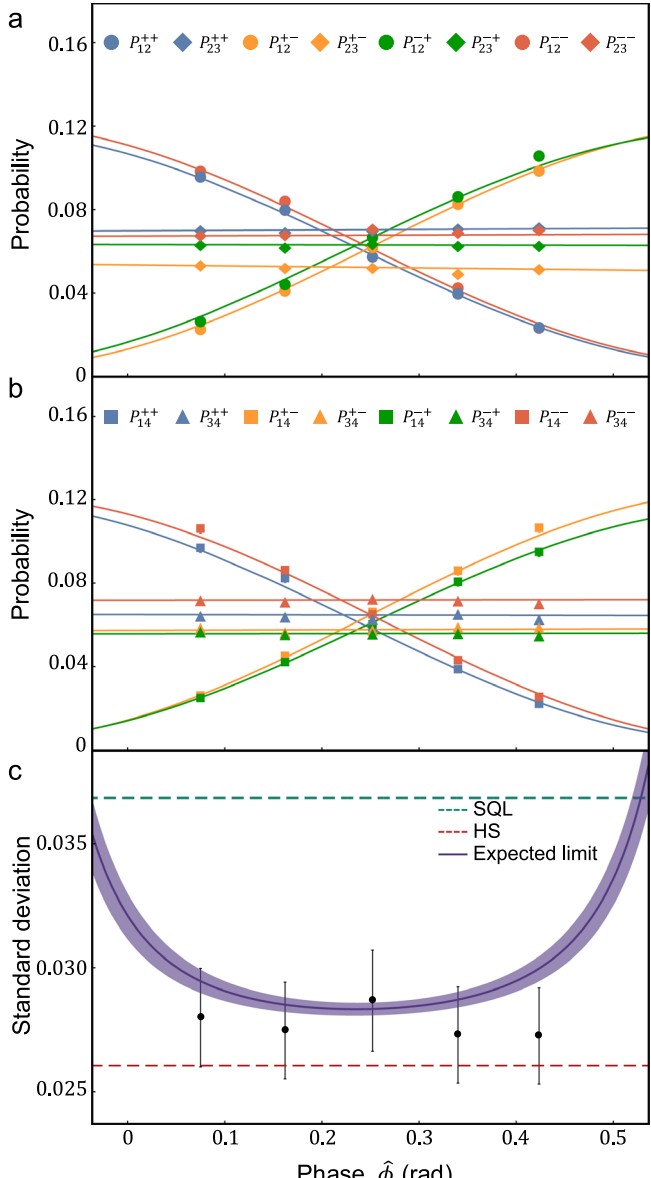

**Fig. 4 | Experimental results on the estimated phase $\hat{\phi}_{est}$ and the standard deviation $\Delta\hat{\phi}_{est}$. a, b** Experimentally obtained interference fringes of sixteen detection probabilities versus $\hat{\phi}$. Error bars are smaller than the markers. **c** The purple solid line represents the corresponding weak form of CRB value. The purple shaded area corresponds to 95% confidence regions, which are obtained from the standard deviation of fitting parameters. Cyan and red dashed lines correspond to the SQL and the HS, respectively. The error bars represent the standard deviation with $s = 100$ groups. See Methods. Detailed information on $\hat{\phi}_{est}$ and $\Delta\hat{\phi}_{est}$ is provided in Supplementary Note.

Then the sensitivity bound can be calculated from Eq. (7) as $\Delta^2\hat{\phi} = 1/N^2$, which is the HS of $N$ photons. We note that the HS of $N$ photons can be described using the average number of photons per mode, denoted as $n = N/d$, i.e., $\Delta^2\hat{\phi} = 1/N^2 = 1/(n^2 \times d^2)$. It is noteworthy that our proposed protocol implies the HS for both the average photon number, $n$, and the number of unknown phases, $d$, which is desirable for distributed quantum sensing[16,19]. Moreover, note that in our scenario, it does not require the photon number $N$ to be equal to or larger than the number of unknown phases $d$. It means that one can estimate the average of $d$ unknown phases with $N$ photons even for $N < d$. See Supplementary Note for detailed calculation on the generalization of our scheme.

## Discussion

In summary, we experimentally demonstrated the distributed quantum phase sensing among four nodes with 2.2 dB sensitivity enhancement over the SQL. In our work, we estimated an average of four unknown phases located in distant nodes, which are separated by 3 km away from the central node. We theoretically showed that in our proposed scheme, one can achieve the HS even when the number of photons is less than the number of unknown phases. Moreover, we proposed a generalization of our scheme for estimating spatially distributed $d$ phases using $N$ photons with sensitivity of the HS. In addition, by utilizing the multiple passes of the phase shifter,

our proposed scheme can estimate a linear global function of the distributed phases with the HS[21,22,34]. We believe that our results can provide helpful guidelines for a practical demonstration of spatially distributed quantum sensing by increasing the number of nodes consisting of the distributed sensor network. In addition, by exploiting the state-of-the-art long-distance entanglement distribution technique through deployed optical fiber networks[35-37], our distributed quantum sensing scheme can be demonstrated in deployed optical fiber networks. It is also intriguing to combine distributed quantum sensing with simultaneous estimation of multiple parameters at a single location[26-29,38-40] by considering a scenario in which one finds an optimal strategy for estimating spatially

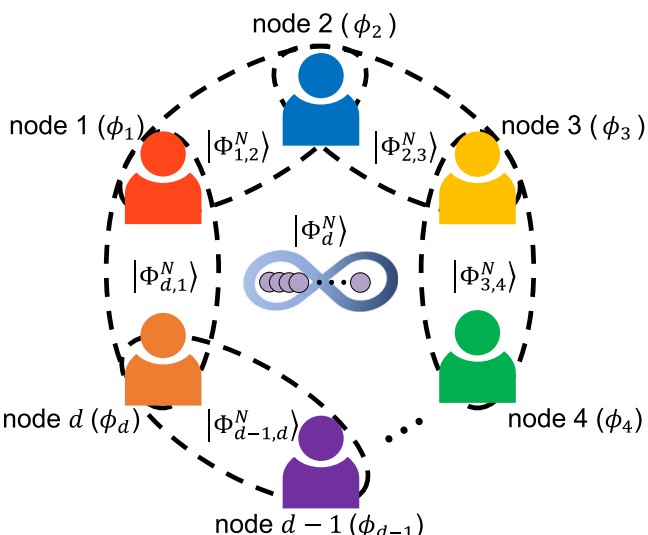

**Fig. 5 | Generalized distributed sensing scheme with the $\left|\Psi_d^N\right\rangle$ states.** Probe state $\left|\Psi_d^N\right\rangle$ are directed to two nodes out of $d$ nodes, which is the superposition of multiphoton two-mode entangled states. The goal is to estimate the linear combination of distributed $d$ unknown phases $\hat{\phi}$.

distributed multiple parameters when each node has more than a single parameter to be estimated.

## Methods

### Theoretical calculation of quantum Fisher information matrix and $\Delta^2\hat{\phi}$

We theoretically analyzed the ideal sensitivity limit of our scheme by calculating the weak form of CRB. By taking into account $\{P_l\}$, one can obtain the sensitivity bound from the Fisher information matrix **F**, which can be calculated as follows:

$$\mathbf{F} = \begin{pmatrix} 1/2 & 1/4 & 0 & 1/4 \\ 1/4 & 1/2 & 1/4 & 0 \\ 0 & 1/4 & 1/2 & 1/4 \\ 1/4 & 0 & 1/4 & 1/2 \end{pmatrix}. \tag{8}$$

Here, we can see the non-zero off-diagonal terms between the two adjacent nodes. Then the uncertainty bound of $\hat{\phi}$ can be calculated to be $\Delta^2\hat{\phi} = 1/4$ by using Eq. (1) with $\boldsymbol{\alpha} = (1/4, 1/4, 1/4, 1/4)$. Note that the sensitivity bound of 1/4 is equal to the HS $1/N^2$ when $N=2$, and here the SQL is defined as $1/N = 1/2$.

### Error analysis

The standard deviation of estimated phase $\hat{\phi}_{\mathrm{est}}$ was obtained from experimentally obtained $\{P_l\}$ with $\mu$ measurements $s$ times using MLE. In our experiments, about 36,700 coincidence counts are divided into 100 groups for each phase shift. The error bar of standard deviation is obtained from an approximation $\delta(\Delta\hat{\phi}) = \Delta\hat{\phi}/\sqrt{2(s-1)}$ with $s = 100$ groups[21,22].

## Data availability

The data that support the findings of this study are available from the corresponding author upon request.

## Code availability

The code used to generate the figures within this paper and other findings of this study are available from the corresponding author upon request.

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

## Acknowledgements

This work was partly supported by National Research Foundation of Korea (NRF) grant funded by the Korea government (MSIT) (2021R1C1C1003625, 2022M3E4A1043330, 2022M3K4A1097123), Institute for Information & communications Technology Planning&Evaluation (IITP) grant funded by the Korea government (MSIT) (2020-0-00947, RS-2023-00222863), the National Research Council of Science and Technology (NST) (CAP21031-200), and the KIST research program (2E32241).

## Author contributions

H.-T.L. initiated and led the project. The experimental scheme was designed by D.-H.K. S.H., Y.-S.K., Y.-K., H.-T.L., and performed by D.-H.K., S.H. Theoretical calculations were carried out by D.-H.K., S.H., S.-W. L., S.-Y.L., C.L., and H.-T.L. D.-H.K., S.H., and H.-T.L. analyzed the experimental data. R.C.P. and K.O. helped to analyze the data and discuss the results. All authors contributed to the discussions and writing the manuscript.

## Competing interests

The authors declare no competing interests.
