## [Peer Review File · Nature Communications]

Distributed quantum sensing of multiple phases with fewer photonsEditorial Note: This manuscript has been previously reviewed at another journal that is not operating a transparent peer review scheme. This document only contains reviewer comments and rebuttal letters for versions considered at *Nature Communications*.

REVIEWERS' COMMENTS

Reviewer #1 (Remarks to the Author):

The authors provided a detailed response to our questions regarding the efficiency of the threshold detection. I have gone through the authors' calculations. It should be noted that Reviewer 3 and Reviewer 4 also raised concerns about the practicality of this study. In the response, the authors further emphasized that their method can achieve the Heisenberg limit as the number of photons increases and can be applicable when the number of parameters to be measured is greater than the number of photons. They also explained in detail the differences between their approach and the alternative solutions mentioned by Reviewer 3 and 4, highlighting the advantages of their method in terms of increasing the number of photons.

In short, the authors provided a comprehensive response to the questions raised by the reviewers and further explained the advantages and potential applications of their method. They clearly expressed the contribution of their research to the field of distributed quantum sensing and emphasized the applicability of their method in different scenarios. I believe it is suitable for publication in *Nature Communications*.

Reviewer #2 (Remarks to the Author):

The authors have significantly improved the previous version of the manuscript. Before final acceptance, however, there are a few points that should be addressed:

-) Everywhere in the text, including in the abstract and figure captions, the word "Heisenberg limit" must be replaced by "Heisenberg scaling".

-) Regarding the Cramer-Rao bound, the word "effective form" must be replaced by "weak bound" (compared to the strong bound obtained with the inverted Fisher matrix).

-) The experiment has been devoted to the estimation of an average of four spatially distributed phases. According to the authors this choice was done for "demonstration purposes only". However, I would like the authors explicitly comment on the possible protocol to estimate with their apparatus different combination of phases. In particular, I believe that different phase combinations will have reduced sensitivity gains with respect to the shot noise. This is a typical scenario, see for instance Phys.

Rev. A 108, 032621 (2023) (that the authors can consider to cite) that does not diminish the value of the results.

Reviewer #3 (Remarks to the Author):

We carefully read through the revised manuscript and response letter. The comments are the following:

1. The revised version add some details to the manuscript, which is good for the readers to better understand the research. The experiment is well implemented and theory is novel to some extent, however, we still think the novelty of the work itself does not meet the standard of Nature Communication.
2. The reply to the comments given by Reviewer 3 is not straightforward. Especially, to take advantage of Heisenberg limit, N need to be large. After all, $1/N^2$ is smaller than $1/N$ only when $N > 1$ and the difference becomes more obvious when N is large. However, when $N < 1$, $1/N^2$ will be larger than $1/N$.
3. The reply to the first comment from Reviewer 2 is not straightforward as well. What we can see is just changing the name of a quantity, but the problem itself is not solved. Though, we do notice some other published papers have the same problem: inverse of a bilinear form matrix is not the same as a bilinear form of the inverse matrix. Of course, not all matrix has an inverse matrix, but for a physical quantity, this problem can be solved at least asymptotically.

Reviewer #4 (Remarks to the Author):

In my review of an earlier version of this manuscript I felt that the manuscript was clear and well-written and presented interesting experimental results demonstrating a new protocol for distributed quantum sensing. I thought it would be a valuable addition to the literature but had concerns over its potential impact. This was mainly due to the fact that it was not clear how the measurement precision scales with the number of nodes, d , nor was it clear whether the scheme would be practical or whether any precision gained would justify the difficulty with implementing the scheme.

I think that these issues have been adequately addressed in the new version and so I am happy to recommend publication.

Reviewer #1 (Remarks to the Author):

The authors provided a detailed response to our questions regarding the efficiency of the threshold detection. I have gone through the authors' calculations. It should be noted that Reviewer 3 and Reviewer 4 also raised concerns about the practicality of this study. In the response, the authors further emphasized that their method can achieve the Heisenberg limit as the number of photons increases and can be applicable when the number of parameters to be measured is greater than the number of photons. They also explained in detail the differences between their approach and the alternative solutions mentioned by Reviewer 3 and 4, highlighting the advantages of their method in terms of increasing the number of photons.

In short, the authors provided a comprehensive response to the questions raised by the reviewers and further explained the advantages and potential applications of their method. They clearly expressed the contribution of their research to the field of distributed quantum sensing and emphasized the applicability of their method in different scenarios. I believe it is suitable for publication in Nature Communications.

→ We thank the Reviewer #1 for reviewing our manuscript and recommending our manuscript to be published in Nature Communications.

Reviewer #2 (Remarks to the Author):

The authors have significantly improved the previous version of the manuscript. Before final acceptance, however, there are a few points that should be addressed:

-) Everywhere in the text, including in the abstract and figure captions, the word "Heisenberg limit" must be replaced by "Heisenberg scaling".

→ We thank the Reviewer #2 for his/her helpful comment. We have modified the "Heisenberg limit" to "Heisenberg scaling".

Manuscript is revised as the following:

a) All "Heisenberg limit" was revised to "Heisenberg scaling" in the revised main text and Supplementary Note.

-) Regarding the Cramer-Rao bound, the word "effective form" must be replaced by "weak bound" (compared to the strong bound obtained with the inverted Fisher matrix).

→ We thank the Reviewer #2 for his/her valuable comment. Following the Reviewer #2's comment, we have revised the "effective form" to "weak form" in the main text and supplementary note.

Manuscript is revised as the following:

a) All "effective form" was revised to "weak form" in the revised main text and Supplementary Note.

-) The experiment has been devoted to the estimation of an average of four spatially distributed phases. According to the authors this choice was done for "demonstration purposes only". However, I would like the authors explicitly comment on the possible protocol to estimate with their apparatus different combination of phases. In particular, I believe that different phase combinations will have reduced sensitivity gains with respect to the shot noise. This is a typical scenario, see for instance Phys. Rev. A 108, 032621 (2023) (that the authors can consider to cite) that does not diminish the value of the results.

⇒ The Heisenberg scaling can be achieved to estimate a linear global function of multiple phases, i.e. $\hat{f} = \sum_{i=1}^M \alpha_i \phi_i$ where M is any integer. It has been both theoretically reported in Ref. [16] and experimentally reported in Ref. [22, 23]. The experimental report in Ref. [22] involved estimating an uneven linear global function of the

distributed phases: $\hat{f} = \sum_{i=1}^M \alpha_i \phi_i$, achieved through multiple phase encoding. In our experimental demonstration, we only consider the case of estimating average of four phases, but, utilizing the same phase encoding technique,

our proposed states can also estimate a linear global function of the distributed phases with achieving the Heisenberg scaling. Following the Reviewer #2's recommendation, we added a sentence in the summary part to mention the possibility of estimating the global linear combination of the distributed phases and cite the recommended reference of Phys. Rev. A **108**, 032621 (2023).

Manuscript is revised as the following:

a) Main text, right column, line 14. We revised the following sentence to explain that a protocol is possible to estimate a global linear combination of the distributed phases.

"In addition, by utilizing the multiple passes of the phase shifter, our proposed scheme can estimate a linear global function of the distributed phases with the Heisenberg scaling."

b) Main text, references. Following reference is added in main text.

"[34] Malitesta, M, Smerzi, A, & Pezz'e, L. Distributed quantum sensing with squeezed-vacuum light in a configurable array of Mach-Zehnder interferometers. Phys. Rev. A. 108, 032621 (2023)."

Reviewer #3 (Remarks to the Author):

We carefully read through the revised manuscript and response letter. The comments are the following:

1. The revised version add some details to the manuscript, which is good for the readers to better understand the research. The experiment is well implemented and theory is novel to some extent, however, we still think the novelty of the work itself does not meet the standard of Nature Communication.

- We thank the Reviewer #3 for his/her valuable comments. We have revised our manuscript and are now confident that our revised manuscript is suitable for publication in Nature Communication as all of the other Reviewers recommended.

2. The reply to the comments given by Reviewer 3 is not straightforward. Especially, to take advantage of Heisenberg limit, N need to be large. After all, $1/N^2$ is smaller than $1/N$ only when $N > 1$ and the difference becomes more obvious when N is large. However, when $N < 1$, $1/N^2$ will be larger than $1/N$.

- In our proposed scheme, the photon number N is the total number of photons used to estimate distributed phases, not the average number of photons at each mode. The N is always larger than 1, and the assumption that it is less than $N < 1$ is not possible in our proposed scheme. In our experimental demonstration, two photons $N=2$ are used, so our protocol can always take advantage of Heisenberg scaling.

3. The reply to the first comment from Reviewer 2 is not straightforward as well. What we can see is just changing the name of a quantity, but the problem itself is not solved. Though, we do notice some other published papers have the same problem: inverse of a bilinear form matrix is not the same as a bilinear form of the inverse matrix. Of course, not all matrix has an inverse matrix, but for a physical quantity, this problem can be solved at least asymptotically.

- We thank the Reviewer #3 for his/her valuable comment. We agree that this problem has not been fully resolved. However, at least, in our work and the published papers [22, 23], the weak bound (effective form of the CRB) gives the exact Heisenberg limit, i.e., $1/N^2$, and they were experimentally achieved. This suggests that, at least in these cases, the effective form of the CRB can serve as a practical sensitivity bound. As pointed out by the Reviewer #3, it will be a meaningful direction to address this issue and is being pursued by authors.

Reviewer #4 (Remarks to the Author):

In my review of an earlier version of this manuscript I felt that the manuscript was clear and well-written and presented interesting experimental results demonstrating a new protocol for distributed quantum sensing. I thought it would be a valuable addition to the literature but had concerns over its potential impact. This was mainly due to

the fact that it was not clear how the measurement precision scales with the number of nodes, d , nor was it clear whether the scheme would be practical or whether any precision gained would justify the difficulty with implementing the scheme.

I think that these issues have been adequately addressed in the new version and so I am happy to recommend publication.

→ We thank the Reviewer #4 for his/her helpful comment and recommending our manuscript to be published in Nature Communications.

List of changes

1. Main text and Supplementary Note. We have modified the term from 'Heisenberg limit' to 'Heisenberg scaling' in response to the comments from the Reviewer #2.
2. Main text and Supplementary Note. We have revised the term from 'effective form' to 'weak form' in accordance with the comments from the Reviewer #2.
3. Main text Page 5, right column, line 14. We revised the following sentence to explain that a protocol can estimate a global linear combination of the distributed phases.

“In addition, by utilizing the multiple passes of the phase shifter, our proposed scheme can estimate a linear global function of the distributed phases with the Heisenberg scaling.”

4. Main text, references. Following reference is added in main text.

“[34] Malitesta, M, Smerzi, A, & Pezz`e, L. Distributed quantum sensing with squeezed-vacuum light in a configurable array of Mach-Zehnder interferometers. Phys. Rev. A. 108, 032621 (2023).”

5. Main text Page 4, Figure 4, caption. We revised the following sentence to explain the error bars. *“Error bars are smaller than markers.”*

6. Main text and Supplementary Note. We modified the “4” to “\$”.

7. Main text and Supplementary Note. We modified the “a” to “a”.

8. Supplementary Note, affiliation. We added an affiliation to match the main text.

“Department of Physics, Korea University, Seoul 02841, Korea”

“Weapon Systems Engineering, ADD School, University of Science and Technology, Daejeon, 34060, Korea”

9. Supplementary Note Page 1, line 4. We revised the following sentence to clarify the experiment setup. *“The polarization of the pump laser is set to $|H\rangle$ using a set of a quarter-wave plate (QWP), a half wave-plate (HWP), and a polarizing beam splitter (PBS).”*

10. Supplementary Note Page 1, line 6. We revised the following sentence to correct the typo.

“After passing through the dual wavelength PBS (DWP) working for both 780 nm and 1560 nm photons, the diagonal polarization of the pump laser is divided by horizontal and vertical polarizations, and the pump laser with $|H\rangle$ ($|V\rangle$) polarization goes through clockwise (counter-clockwise) direction inside a Sagnac interferometer.”

11. Supplementary Note Page 4, equation (5). We revised “U” to correct the typo.